# Threshold Responses in the Taxonomic and Functional Structure of Fish Assemblages to Land Use and Water Quality: A Case Study from the Taizi River

**Yuan Zhang [1], Xiao-Ning Wang [2], Hai-Yu Ding [3], Yang Dai [1], Sen Ding [1],\* and Xin Gao [1]**

[1] State Key Laboratory of Environmental Criteria and Risk Assessment, Chinese Research Academy of Environmental Sciences, Beijing 100012, China; zhangyuan@craes.org.cn (Y.Z.); daiyang815@126.com (Y.D.); gaoxin@craes.org.cn (X.G.)

[2] College of Fisheries, Huazhong Agriculture University, Wuhan 430079, China; xiaoninghzau@163.com

[3] Department of Ecology and Ecosystem Management, Technische Universitaet Muenchen, 85354 Freising-Weihenstephan, Germany; xiaoninghzau@163.com

\* Correspondence: bearnaise@163.com

**Abstract:** Biological functional traits help to understand specific stressors that are ignored in taxonomic data analysis. A combination of biological functional traits and taxonomic data is helpful in determining specific stressors which are of significance for fish conservation and river basin management. In the current study, the Taizi River was used as a case study to understand the relationships between the taxonomic and functional structure of fish and land use and water quality, in addition to determining the thresholds of these stressors. The results showed that taxonomic structure was significantly affected by the proportion of urban land and specific conductivity levels, while functional metrics were influenced by the proportions of farmland and forest. Threshold indicator taxa analysis found that *Phoxinus lagowskii*, *Barbatula barbatula nuda*, *Odontobutis obscura*, and *Cobitis granoei* had negative threshold responses along the gradients of urban developments and specific conductivity. There was a significant change in fish taxonomic composition when the proportion of urban land exceeded a threshold of 2.6–3.1%, or specific conductivity exceeded a threshold of 369.5–484.5 μS/cm. Three functional features—habitat preference, tolerance to disturbances, and spawning traits—showed threshold responses to the proportion of farmland and forest. The abundance of sensitive species should be monitored as part of watershed management, as sensitive species exhibit an earlier and stronger response to stressors than other functional metrics. Sensitive species had a positive threshold response to the proportion of forest at 80.1%. These species exhibited a negative threshold response to the proportion of farmland at 13.3%. The results of the current study suggest that the taxonomic and functional structure of fish assemblages are affected by land use and water quality. These parameters should be integrated into routine monitoring for fish conservation and river basin management in the Taizi River. In addition, corresponding measures for improving river habitat and water quality should be implemented according to the thresholds of these parameters.

**Keywords:** stream fish; land use; water quality; threshold; taxonomic and functional structure

## 1. Introduction

A multi-scale conceptual framework illustrates the spatial organization of stream ecosystems [1], and most ecologists have realized that stressors at different spatial scales (e.g., catchment scale, local scale) are strongly associated with changes in stream fish communities [2,3]. Changes in land use at the catchment scale directly influence water quality, channel stability and riparian habitat quality and affect important ecological processes such as hydrological regimes, primary productivity, metabolism

and organic matter turnover [4,5]. These changes can directly or indirectly affect fish assemblages, potentially causing detrimental effects [6]. Therefore, land use should be considered as a catchment scale environmental driver in river basin management. Management systems which integrate water and land resources are now widely accepted by watershed managers [7,8], whose core concerns are habitat improvement, local ecological restoration, and land use optimization at the catchment scale. Therefore, from a scientific perspective, it is crucial to identify specific stress factors and their associated thresholds, above or below which biological conditions may significantly change.

Understanding the relationship between fish communities and stressors is widely accepted as the basis for fish conservation [9–11]. Several studies have used species taxa or abundance data to examine biotic responses to anthropogenic drivers [12,13]; however the data collected in these studies did not allow for the identification of specific drivers affecting fish function, e.g., feeding habits and spawning behaviors [14]. For example, Brosse et al. [15] found that the descriptors of fish taxonomic structure such as species richness and biomass were not sensitive to mining activities, whereas fish functional structure was significantly affected by mining activities, favoring ubiquitous species at the expense of specialist species. Biodiversity losses are not only changes in abundance and richness, but also changes in trophic positions, rarity, and environmental specialization, as various fish species are not likely to equally contribute to ecosystem function [16]. Thus, a functional approach will help watershed managers to identify specific drivers and plan for the mitigation of potential risks [17–19]. Although functional traits have been used to understand the response mechanism of a fish assemblage to specific stressors [20–22], the quantitative relationship between functional structure and human stressors is not well known.

Studies of environmental thresholds have become increasingly important considering the global decline of freshwater fish diversity. These studies help in understanding the critical management requirements of human disturbances [23]. In previous studies, many methods were used to identify environmental thresholds, such as piecewise regression [24], continuous response functions [25], Bayesian change-point models [26] and nonparametric deviance reduction [27]. These statistical methods are limited as they do not incorporate species-specific responses to the anthropogenic drivers [28] when multivariate species data are processed [26]. Baker and King [29] proposed the Threshold Indicator Taxa ANalysis (TITAN) method to screen significant species from a biological assemblage and determine the tipping points at species and assemblage-levels along the environmental gradient. Furthermore, this method has been applied in environmental threshold identification for many groups, including algae [30], macroinvertebrates [31], macrophytes [32], fish [33], and birds [34]. These tipping points were determined using biotic taxonomic structure data and the TITAN method. In this paper, the TITAN method was combined with a functional approach, to identify the threshold responses of fish functional metrics to specific stressors in the Taizi River.

The Taizi River was selected as the study area, because the landscape patterns and water quality conditions in this area are highly variable. These variations impact the aquatic community, especially fish assemblages [35]. An understanding of stressors and their management requirements at specific scales is required for effective fish conservation. The first aim of this study was to investigate the response of fish assemblages to environmental factors from taxonomic and functional perspectives at the catchment and local scales. It was hypothesized that functional and taxonomic approaches could identify different stressors at different spatial scales. The second aim was to determine the response thresholds of fish assemblages to species-level and assemblage-level stressors. The results of this study will provide important baseline information for setting management objectives for Taizi River catchment management.

## 2. Methods

### 2.1. Study Area and Sampling Sites

The Taizi River catchment is in Liaoning Province, northeastern China. It occurs within a warm temperate zone and has a humid and sub-humid climate. The Taizi River flows from the southeastern mountains to the northwestern plain. The length of the Taizi River is 413 km and it has a catchment area of approximately 13,880 km$^2$. This catchment has undergone dramatic changes in land use patterns, water quality, and fish distribution over the last three decades. From 2000 to 2010, the urban area in the catchment increased by 25.8% [36], in line with recent rapid urbanization across China. Land use pattern in the riparian zone of the Taizi River catchment has shown obvious changes along the longitudinal gradient of the river. The upper region is classified as a mountainous forest region consisting mainly of broadleaf deciduous forest. The middle and lower reaches are in hilly and plain regions, where there is large-scale agricultural development and several important industrial cities, such as Anshan city and Liaoyang city [11]. Our unpublished data illustrated that in the 1980s, the water quality of the Taizi River was above standard, class III according to the environmental quality standards for surface water [37]. However, water quality data from 2016, provided by the local government, indicated that water quality had declined to below class V in the middle and lower reaches of the Taizi River. These changes have resulted in a serious degradation in aquatic ecosystems [11,35]. In comparison with historical records from the 1980s [38], only 44 fish species previously recorded were collected during annual river ecosystem monitoring since 2009, while 62 fish species that were previously recorded could not be found during this routine monitoring. Although local watershed managers have increasingly focused on stream ecosystem health [8], it is still unclear how to conserve or restore aquatic species populations, given the poor understanding of the relationships between biotic assemblages and stressors at local and catchment scales.

A total of 53 sites were sampled in the Taizi River catchment between August 2009 and October 2010. Twenty sampling sites were located on the main stream and 33 sampling sites were located on tributaries (Figure 1). At each site, data regarding physico-chemical water properties, land use, and fish assemblages were collected according to the following methods.

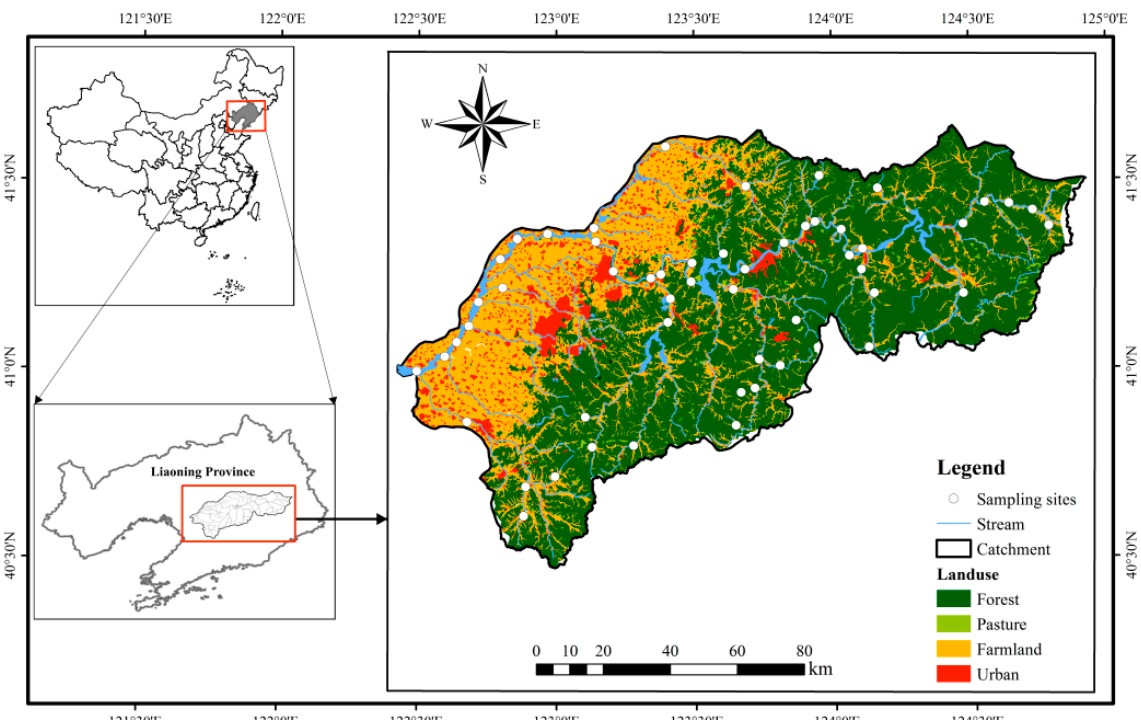

**Figure 1.** Study sites in the Taizi River catchment.

## 2.2. Water Physico-Chemical Conditions

Water temperature (WT, °C), pH, dissolved oxygen concentration (DO, mg/L), specific conductivity (SC, µS/cm), total dissolved solids (TDS, mg/L), suspended solids (SS, mg/L), total nitrogen (TN, mg/L), total phosphorus (TP, mg/L), five-day biochemical oxygen demand ($BOD_5$, mg/L), chemical oxygen demand ($COD_{Mn}$, mg/L) and ammonium nitrogen ($NH_4^+$, mg/L) were measured at each sampling site. WT, pH, DO, SC and TDS were measured using the YSI Pro 2030 multi-parameter meter (USA). At each sampling site, 500 mL of water was collected from the left, middle, and right channel. These three water samples were mixed into a 2 L bottle and stored in a low-temperature sample box. Water samples were immediately transported to the laboratory, where TN, TP, $COD_{Mn}$, $NH_4^+$ and $BOD_5$ were measured. All water parameters were determined according to methods specified in the environmental quality standards for surface water [39]. In the field, 300 ml of stream water was passed through a vacuum filter, with a 0.45 µm filter membrane. The initial weight of the filter membrane was subtracted from the weight of the filter membrane after drying in the laboratory to calculate SS.

## 2.3. Land Use Patterns at the Catchment Scale

Land use data for the Taizi River were attained from SPOT5 (Satellite Positioning and Tracking 5) image data from September 2007, at a resolution of 2.5 m. The multi-spectral and panchromatic bands were subject to geometric correction and Albers projection, and were then fused by ERDAS (Earth Resources Data Analysis System). Land use patterns were interpreted and obtained from the fused map. It has been well documented that urban, farmland, forest and pasture land use have significant effects on stream fish assemblages [6], and therefore these four land use types were used as independent variables to analyze the relationship between landscape and fish assemblage. The proportion of forest, pasture, farmland, and urban land was calculated in ArcGIS 10.2, in addition to the scope of drainage area for each site (Figure 1 and Table A1).

## 2.4. Fish Assemblages

At each sampling site, a 300 m survey reach containing different types of micro-habitats was delimited for fish collection. In areas that could be accessed by wading, fish were sampled by electrofishing for 30 min following a zig-zag path within the survey reach [40]. In areas that were too deep for wading, electrofishing was carried out from an inflatable boat. Two gill nets with a mesh size of $3 \times 3$ cm and $6 \times 6$ cm, respectively, were used for sampling. The gill nets were set up at the lower end of each sampling reach, with a one-hour residence time. Once collected, fish were identified at the species level and biological information recorded.

Following a literature review, four life history and habitat preference parameters for fish species were chosen [38]. Fish exhibited two habitat preferences which allowed them to be classified as either pelagic or demersal species. Based on trophic guild, four types of fish species were identified: piscivores, omnivores, herbivores, and detritivores. Sensitive and tolerant species were classified by the response of fish species to environmental disturbances. Spawning traits were used to divide fish species into three groups: pelagic-egg species, demersal-egg species, and viscid-egg species.

## 2.5. Data Analysis

A Spearman correlation two-tailed test was used to explore the relationship between land use patterns at the catchment scale and physico-chemical water conditions at the local scale. The significance level of the correlation of paired variables ($p < 0.05$) was calibrated by the Bonferroni test [41].

Partial Canonical Correspondence Analysis (pCCA) was used to explore the relationships between the taxonomic and functional structures of fish assemblages and stressors (i.e., water quality and land use patterns at the catchment level). River order was used as a covariate and anthropogenic drivers were used as explanatory variables to account for the longitudinal gradient of fish community with drainage area. Prior to analysis, any rare species which had been recorded once during sampling

were excluded, as shown in Table 1. Species abundance data were $\log_{10}$ (x + 1) transformed to reduce the variation in abundance. A preliminary forward selection pCCA with all stressors was used to identify and delete redundant variables, for example, those with a high variance inflation factor (>20) [11]. Forward selection and Monte Carlo permutation analysis with 499 permutations were used to select a minimum set of stressors which had significant and independent effects on fish assemblage structure [42]. The significance level was set at $p < 0.05$. pCCA was performed using CANOCO software for Windows Version 4.5 (Biometris, Wageningen and Petr Smilauer, Ceske Budejovice).

　　　TITAN, which is an analytic method that combines and extends the change-point analysis and indicator species analysis, was used to identify individual and cumulative fish species responses to physico-chemical conditions and land use patterns [29]. TITAN examined the value of a predictor variable that maximizes association of individual species with one side of the partition, either positively or negatively. Association was measured by the indicator value (IndVal) [43]. Bootstrapping was used to identify significant indicator species [29]. The response direction of individual species to the pressure gradient, which is either positive or negative, was confirmed if the following conditions were met: the change in frequency and abundance of the species was in the same direction for more than 95% of 500 bootstrapped runs (i.e., high purity) and at least 95% of 500 bootstrapped runs were significantly different ($p < 0.05$) from randomly distributed data (i.e., high reliability). The sum of IndVal z-scores was used to examine the thresholds and confidence limits of fish assemblages. Peaks in the sum of z-scores along the pressure gradient were associated with the maximum decline among negative responders ($z-$) or maximum increase among positive responders ($z+$) of frequency and abundance. Prior to TITAN analysis, any species which occurred at less than five sites were excluded to remove outliers representing a potential bias (Table 1). The threshold responses of fish functional structure were analyzed at the species level, with the aim of identifying significant indicating metrics and their tipping points. TITAN analysis was performed using R software [44] using the package "TITAN", following the method developed by Baker and King [29].

**Table 1.** Species composition, abundance and ecological characteristics of fish species recorded in the Taizi River catchment (* species not used for TITAN analysis as they were recorded at less than five sites; # species excluded from pCCA as they were only recorded at a single site. Deme = demersal, Pela = pelagic. Pisc = piscivorous, Omni = omnivorous, Herb = herbivorous, Detr = detritivorous. Sens = sensitive, Tole = tolerant. Peeg = pelagic-egg, Deeg = demersal-egg, Vieg = viscid-egg).

| Family | Species (Scientific Name) | Code | Individual Numbers | Occurring Frequency | Preferential Environment | Trophic Status | Environmental Response | Spawning Trait |
|---|---|---|---|---|---|---|---|---|
| Petromyzonidae | *Lampetra mori* *# | Lamp_mo | 1 | 1.9% | Deme | Pisc | Sens | Vieg |
| Cyprinidae | *Zacco platypus* | Zacc_pl | 380 | 62.3% | Pela | Herb | Tole | Deeg |
| | *Leuciscus waleckii* * | Leuc_wa | 22 | 3.8% | Deme | Omni | Sens | Vieg |
| | *Phoxinus lagowskii* | Phox_la | 1873 | 62.3% | Deme | Omni | Sens | Vieg |
| | *Hemiculter leucisculus* | Hemi_le | 27 | 15.1% | Pela | Omni | Tole | Vieg |
| | *Acheilognathus macropterus* *# | Ache_ma | 25 | 1.9% | Deme | Herb | Tole | Deeg |
| | *Acheilognathus chankaensis* | Ache_ch | 117 | 20.8% | Deme | Herb | Tole | Deeg |
| | *Rhodeus lighti* | Rhod_li | 38 | 11.3% | Deme | Omni | Tole | Deeg |
| | *Pseudorasbora parva* | Pseu_pa | 198 | 43.4% | Deme | Omni | Tole | Vieg |
| | *Gobio lingyuanesis* | Gobi_li | 165 | 11.3% | Deme | Detr | Sens | Vieg |
| | *Gobio cynocephalus* | Gobi_cy | 87 | 15.1% | Deme | Detr | Sens | Vieg |
| | *Gobio rivuloides* | Gobi_ri | 8 | 9.4% | Deme | Detr | Tole | Vieg |
| | *Squalidus chankaensis* * | Squa_ch | 14 | 5.7% | Deme | Omni | Sens | Vieg |
| | *Squalidus argentatus* * | Squa_ar | 2 | 3.8% | Deme | Omni | Sens | Vieg |
| | *Huigobio chinssuensis* * | Huig_ch | 4 | 5.7% | Deme | Omni | Tole | Vieg |
| | *Abbottina rivularis* | Abbo_ri | 275 | 54.7% | Deme | Omni | Tole | Deeg |
| | *Abbottina liaoningensis* | Abbo_li | 53 | 22.6% | Deme | Omni | Tole | Deeg |
| | *Pseudogobio vaillanti* * | Pseu_va | 25 | 5.7% | Deme | Detr | Tole | Vieg |
| | *Carassius auratus* | Cara_au | 152 | 47.2% | Deme | Omni | Tole | Vieg |
| Cobitidae | *Barbatula barbatula nuda* | Barb_nu | 1052 | 60.4% | Deme | Omni | Tole | Vieg |
| | *Cobitis granoei* | Cobi_gr | 131 | 50.9% | Deme | Herb | Sens | Vieg |
| | *Lefua costata* * | Lefu_co | 8 | 7.5% | Deme | Omni | Sens | Vieg |
| | *Misgurnus anguillicaudatus* | Misg_an | 83 | 43.4% | Deme | Omni | Tole | Vieg |
| Hemiramphidae | *Hyporhamphus sajori* *# | Hypo_sa | 30 | 1.9% | Pela | Herb | Sens | Vieg |
| Oryziidae | *Oryzias latipes* * | Oryz_la | 5 | 5.7% | Pela | Detr | Tole | Peeg |
| Gasterosteidae | *Pungitius pungitius* * | Pung_pu | 3 | 3.8% | Pela | Detr | Sens | Peeg |
| Cichlidae | *Oreochromis niloticus* *# | Oreo_ni | 3 | 1.9% | Deme | Omni | Tole | Vieg |
| Eleotridae | *Odontobutis obscura* | Odon_ob | 87 | 22.6% | Deme | Detr | Sens | Vieg |
| | *Hypseleotris swinhonis* * | Hyps_sw | 2 | 3.8% | Deme | Pisc | Tole | Vieg |
| Gobiidae | *Rhinogobius brunneus* | Rhin_br | 467 | 35.8% | Deme | Omni | Tole | Vieg |
| | *Tridentiger obscurus* *# | Trid_ob | 5 | 1.9% | Deme | Omni | Tole | Vieg |

## 3. Results

### 3.1. General Patterns of Fish Assemblage

A total of 5342 individual fish, representing 31 species within nine families, were collected from all study sites (Table 1). The dominant species in the study area were *Phoxinus lagowskii* and *Barbatula nuda* whose relative abundances were 35% and 20%, respectively. Cyprinidae was the dominant family, with 18 species recorded during sampling. A single species was recorded for five families: Petromyzonidae, Hemiramphidae, Oryziidae, Gasterosteidae, and Cichlidae. *Zacco platypus*, *P. lagowskii* and *B. barbatula nuda* were ubiquitous and their occurrence frequencies were greater than 60%.

### 3.2. Relationship between Stressors and Fish Assemblages

Forest land was a main land use type in the study area. The proportion of forested land varied from 23% to 92% among sampling sites (Table A1). The upper mountainous region had large areas of forest, whereas the lower plain had a very small proportion of forests due to intensive human activities. Pasture accounted for 0–4% of land use. Farmland and urban land were mainly located at the middle and lower regions of the catchment. The proportion of farmland ranged from 6% to 60%, while urban land accounted for 0% to 21%.

Seven water quality parameters (WT, SC, TDS, SS, TN, $COD_{Mn}$ and $NH_4^+$) showed significant correlations with land use patterns (Table 2). A negative correlation indicated that higher proportions of forested land was favorable for maintaining good water quality, whereas a positive correlation indicated that increasing proportions of farmland and urban land resulted in an increase of WT, SC, TDS, SS, TN, $COD_{Mn}$ and $NH_4^+$ and a degradation in water quality.

**Table 2.** The Spearman's correlation coefficients between land use types and local water quality conditions. (* ($p < 0.05$), ** ($p < 0.01$), or *** ($p < 0.001$) indicates the significance level as performed by the Bonferroni test).

| Water Quality Conditions | Forest | Pasture | Farmland | Urban |
|---|---|---|---|---|
| Water temperature | −0.46 ** | 0.128 | 0.440 ** | 0.356 ** |
| pH | 0.084 | 0.044 | −0.124 | −0.033 |
| Dissolved oxygen concentration | 0.277 | 0.010 | −0.219 | −0.373 |
| Specific conductivity | −0.661 *** | 0.189 | 0.686 *** | 0.533 *** |
| Total dissolved solids | −0.478 *** | 0.181 | 0.447 ** | 0.505 *** |
| Suspended solids | −0.488 *** | 0.122 | 0.466 *** | 0.512 *** |
| Total nitrogen | −0.461 ** | 0.225 | 0.463 *** | 0.352 |
| Total phosphorus | −0.323 | 0.110 | 0.246 | 0.344 |
| Five-day biochemical oxygen demand | −0.230 | 0.064 | 0.265 | 0.219 |
| Chemical oxygen demand | −0.404 ** | 0.107 | 0.389 ** | 0.475 *** |
| Ammonium nitrogen | −0.679 *** | 0.234 | 0.674 *** | 0.558 *** |

The Monte Carlo unrestricted permutation test indicated that fish taxonomic structure and functional structure were significantly affected by different stressors (Tables 3 and 4). Specific conductivity ($p = 0.002$) and urban land use ($p = 0.018$) represented the dominant anthropogenic drivers affecting fish taxonomic structure in the study area, whereas functional structure was significantly affected by the proportion of forest ($p = 0.002$) and farmland ($p = 0.048$). Figure 2 summarizes the spatial trends of stressors in relation to fish species and functional metrics. Species-environmental correlations were high for the first two ordination axes as shown in Figure 2a and Table 3 ($r = 0.914$ for axis 1 and $r = 0.754$ for axis 2) and in Figure 2b and Table 4 ($r = 0.725$ for axis 1 and $r = 0.643$ for axis 2). The first two axes explained 48.3% (axis 1 = 30.9%) of the cumulative variance of species and stressors from the species perspective as shown in Figure 2a and Table 3 and 66.4% (axis 1 = 48.9%) from the functional structure perspective as shown in Figure 2b and Table 4, respectively.

There was notable spatial differentiation between species groups (Figure 2a). Species located at the negative end of the environmental gradient (e.g., *P. lagowskii*, *Odontobutis obscura*, *Cobitis granoei*), generally occurred in the hilly or mountainous reaches of the catchment, where specific conductivity values were low and there was a low proportion of urban land. *Carassius auratus*, *Acheilognathus chankaensis*, *Hemiculter leucisculus*, *Oryzias latipes* and *Gobio rivuloides* were located at the positive end of the environmental gradient (Figure 2a), and these species were more abundant in the plains, with higher specific conductivity levels and proportions of urban land. Trophic guild and spawning traits also showed obvious spatial differentiation along anthropogenic drivers (Figure 2b). There was a positive correlation between the proportion of omnivores (M4) and demersal-egg species (M10) with the proportion of forested land, while these species showed a negative correlation with the proportion of farmland. The proportion of piscivores (M3), pelagic-egg species (M9) and viscid-egg species (M11), increased with increasing farmland cover, and decreasing forested land. Demersal-egg species was found to reach maximum abundance at sites with a low proportion of farmland, as shown in Figure 2b.

**Table 3.** Monte Carlo unrestricted permutation test of stressors and fish taxonomic structure in pCCA.

| Stressors | Axis 1 | Axis 2 | F-Values of Monte Carlo Test | *p*-Value |
|---|---|---|---|---|
| Specific conductivity (μS/cm) | 0.864 | 0.597 | 4.97 | 0.002 |
| Urban (%) | 0.665 | 0.481 | 2.05 | 0.018 |
| Eigenvalues | 0.400 | 0.224 | | |
| Species-environment correlations | 0.914 | 0.754 | | |
| Cumulative percentage variance of species-environment relation | 30.9% | 48.3% | | |

**Table 4.** Monte Carlo unrestricted permutation test of stressors and fish functional structure in pCCA.

| Stressors | Axis 1 | Axis 2 | F-Values of Monte Carlo Test | *p*-Value |
|---|---|---|---|---|
| Forest (%) | −0.816 | 0.092 | 7.44 | 0.002 |
| Farmland (%) | 0.766 | −0.165 | 1.77 | 0.048 |
| Eigenvalues | 0.063 | 0.022 | | |
| Species-environment correlations | 0.725 | 0.643 | | |
| Cumulative percentage variance of species-environment relation | 48.9% | 66.4% | | |

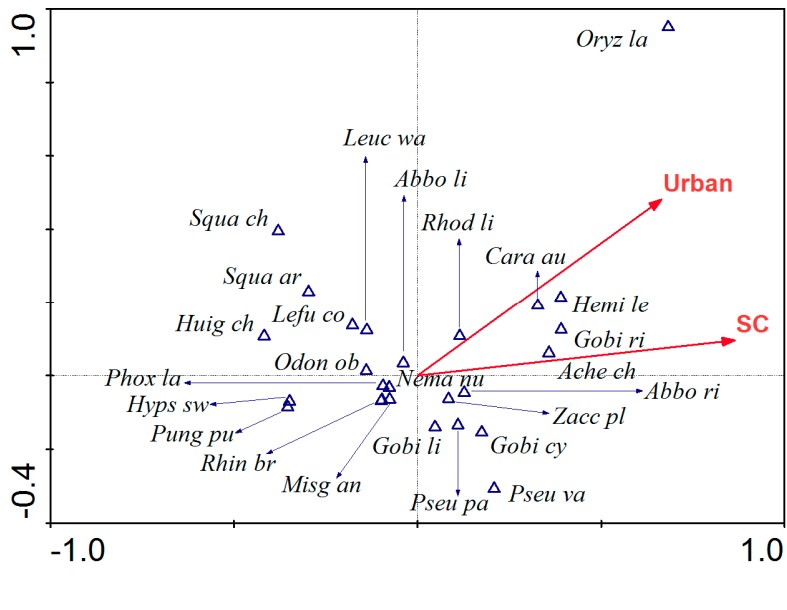

(**a**)

**Figure 2.** *Cont.*

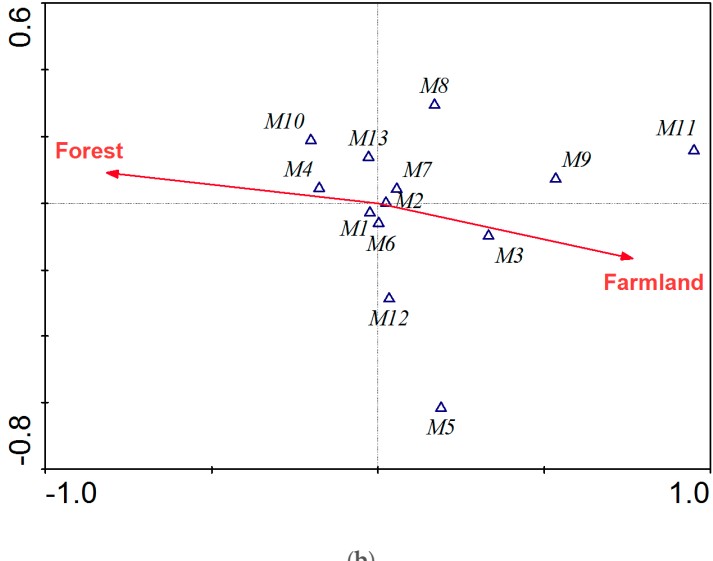

(**b**)

**Figure 2.** pCCA ordination diagram of stressors and taxonomic structure (**a**) (SC: Specific conductivity) and functional structure (**b**) of fish assemblages in the Taizi River catchment (M1: Proportion pelagic species; M2: Proportion of benthic species; M3: Proportion of piscivores; M4: Proportion of omnivores; M5: Proportion of herbivores; M6: Proportion of detritivores; M7: Proportion of as tolerant species; M8: Proportion of sensitive species; M9: Proportion of pelagic-egg species; M10: Proportion of demersal-egg species; M11: Proportion of viscidegg-species).

*3.3. Species and Assemblage Thresholds for Stressor Variables*

TITAN was used to identify the response thresholds of the stressors affecting fish taxonomic and functional structure in pCCA at both fish species level and the assemblage level. The threshold response of species to stressors is shown in Figure 3. *C. granoei* (change point of urban proportion = 1.28%), *B. nuda* (1.53%), *P. lagowskii* (1.77%), *O. obscura* (1.77%) and *Misgurnus anguillicaudatus* (2.62%), were negative indicator species (sensitive species, *z*−) along the gradient of proportion of urban land, whereas *C. auratus* (2.25%), *A. chankaensis* (2.61%) and *Hemiculter leucisculus* (3.14%) were identified as positive indicator species (tolerant species, *z*+) by TITAN (Figure 3a). The assemblage-level threshold of urban proportion was 2.62% for negative indicator species (*z*−) which exhibited a higher sensitivity and 3.14% for positive indicator species (*z*+) which showed a higher tolerance (Table 5).

Four sensitive species (*z*−), *B. barbatula nuda* (280.5 µS/cm), *P. lagowskii* (339 µS/cm), *O. obscura* (362 µS/cm) and *C. granoei* (502.5 µS/cm), and two tolerant species (*z*+), *C. auratus* (484.5 µS/cm) and *A. chankaensis* (523.5 µS/cm), were identified along the gradient of specific conductivity by TITAN as shown in Figure 3b. The assemblage-level threshold of specific conductivity was 369.5 µS/cm for sensitive species (*z*−) and 484.5 µS/cm for tolerant species (*z*+) (Table 5).

Three fish functional features, habitat preference, tolerance to environmental disturbances and spawning traits, showed threshold responses to farmland and forest cover at the catchment scale (Figure 4). The proportion of omnivores (M2), sensitive species (M8) and viscid-egg species (M11) were negative indicator metrics (*z*−) along the gradient of proportion of farmland proportion, but positive indicator metrics (*z*+) with decreasing forest proportion. Additionally, the proportion of pelagic species (M1), tolerant species (M7) and pelagic-egg species (M9), responded positively (*z*+) to farmland cover, and negatively (*z*−) to proportion of forest cover.

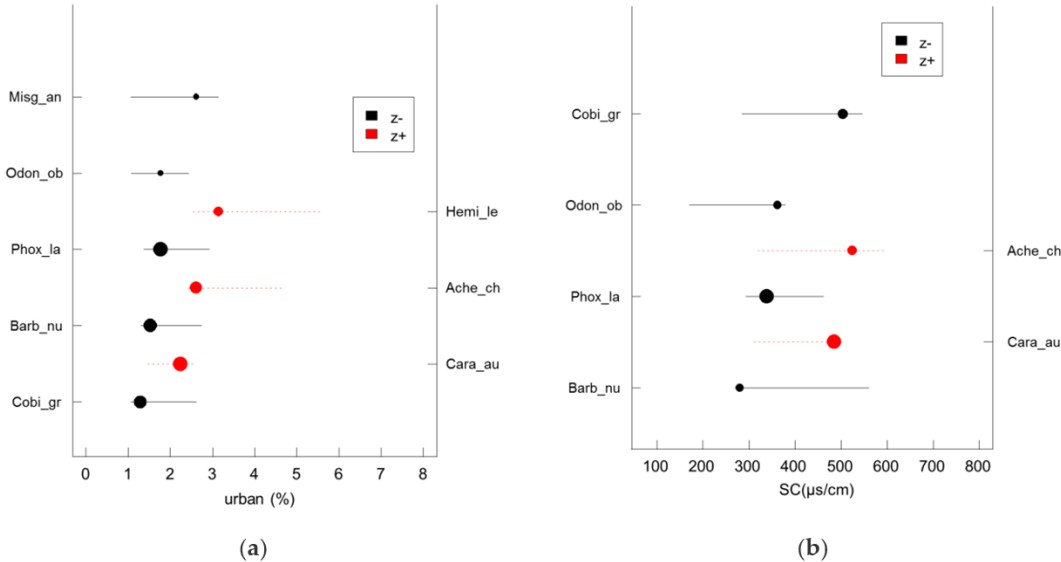

**Figure 3.** The responses and thresholds of fish taxonomic structure to urban land use (**a**) in SC (**b**) in TITAN. Red circles (positive response) represent fish species that become more frequent or abundant along the gradient, while the black circles (negative response) indicate fish species that decline in frequency or abundance. Horizontal lines overlapping each circle indicate the 5th and 95th percentiles of 500 bootstrapped runs. Circles are sized in proportion to the magnitude of the response (z-scores).

**Table 5.** The assemblage-level thresholds of fish taxonomic structure to stressors in the Taizi River. Associated percentiles are based on the frequency distribution of thresholds from 500 bootstrap replicates.

| Stressors | Response Types | Thresholds | 0.05 | 0.50 | 0.95 |
|---|---|---|---|---|---|
| Urban (%) | negative response | 2.62 | 1.28 | 2.42 | 4 |
| | positive response | 3.14 | 1.36 | 2.60 | 5.62 |
| Specific conductivity (μS/cm) | negative response | 369.5 | 218 | 369.5 | 560 |
| | positive response | 484.5 | 300.5 | 450 | 639 |

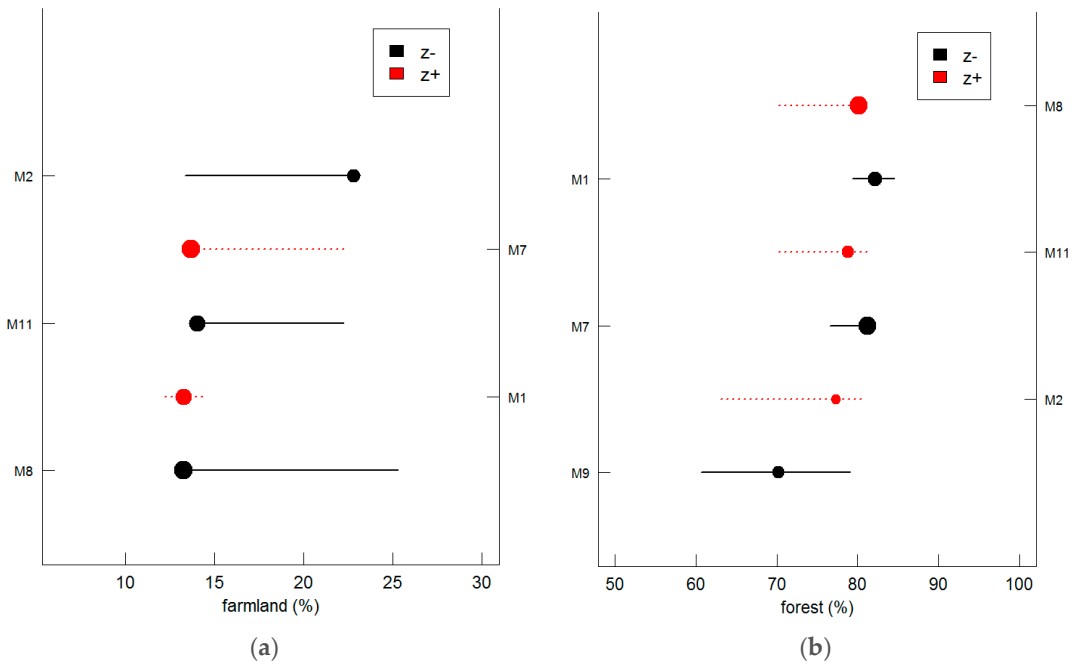

**Figure 4.** The responses and thresholds of fish functional structure metrics to farmland cover (**a**) and forest cover (**b**) in TITAN.

The thresholds of negative indicator metrics ($z-$) were identified as M8 = 13.28%, M11 = 14.04% and M2 = 22.81% along the gradient of farmland cover, whereas the thresholds of positive indicator metrics ($z+$) was identified as M1 = 13.28% and M7 = 13.71% (Figure 4a). The identified change point of forest proportion was 70.18%, 81.21% and 82.08% for M9, M7, and M1, respectively, which were sensitive responders ($z-$) and 77.27%, 78.79% and 80.1% for M2, M11, and M8, respectively, which were tolerant responders ($z+$) (Figure 4b). M8 showed a more sensitive threshold response to forest cover and farmland cover than other functional metrics. The non-parameter test verified there was a significant difference in M8 when these drivers were on both sides of the environmental change point (Figure 5).

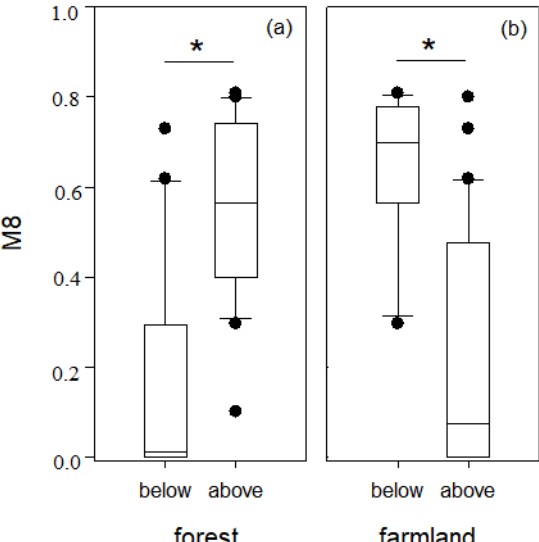

**Figure 5.** Boxplots of M8 (proportion of sensitive species) for the sites below and above the environmental change point of forest cover (**a**) and farmland cover (**b**) (* represents a significant difference as derived by the Mann-Whitney non-parameter test. The horizontal line in each box represents the median value. Solid points indicate extreme abnormal values).

## 4. Discussion

Our results show a nested relationship among two scale stressors and fish assemblages. This indicates that integrated management of land and water is essential for fish conservation. Changes in land use are generally driven by human activities [45], and the conversion from forest or pasture to farmland or urban settlements has strongly affected water quality and biodiversity [6]. Land use development is known to degrade water quality [46]. The results of the current study found a good correlation between most water quality parameters and landscape factors, except for pasture. This is because a very low proportion of pasture in the Taizi River catchment occurs in proximity to stream ecosystems [47]. Landscape changes such as agricultural production, urban development or forest logging contribute to an increase in specific conductivity in streams [48,49]. In the current study, pCCA results revealed three landscape factors (forest, farmland and urban) significantly affecting fish assemblages in the Taizi River. Similar effects on fish assemblages due to deforestation, agricultural development, and rapid urbanization have also been reported by Wang et al. [9] and Pinto et al. [47].

The results of this study support the hypothesis that fish taxonomic and functional structures are affected by stressors. Taxonomic structure was influenced by the proportion of urban land and specific conductivity values, whereas functional structure showed a response to proportion of farmland and forest. The differences in response in taxonomic and functional structure have also been reported by other authors [15,19,50]. Teresa and Casatti found that deforestation influenced fish functional and taxonomic diversity, while physical habitats resulted in changes of functional composition [50]. For example, under severe disturbance, a stream ecosystem will experience a loss in fish species richness,

whereas a less severe disturbance may cause one species to replace another, resulting in a functional change [15]. The proportion of urban land use in the Taizi River catchment increased rapidly from 1.5% in 1980 to 8.2% in 2009 (unpublished data), while concentrations of major ions in the Taizi River have increased several-fold (Table A2), indicating an increase in specific conductivity with the concomitant development of local industry [49]. The strong response exhibited in fish assemblages linked with changes in urban land use and specific conductivity, are likely because many toxic materials originate from urban sewage. These ions result in a direct or cumulative stress on fish physiology, causing a population decline or the loss of sensitive species [49,51]. Hitt and Chambers [52] also found a clear change in the taxonomic structure of fish in streams (i.e., fewer species, lower abundance and biomass) under conductivity disturbance, but there was no change in functional structure. Several previous studies have focused on function-based approaches for identifying biotic responses along a gradient of stressors [18,53], because the functional structure is an important index for studying the function and stability of ecosystems [54]. Functional metrics, such as species richness, biomass, or abundance of organisms within a taxonomic assemblage are also very sensitive to environmental disturbances. The functional metrics in the current study have provided a complementary understanding of stressors that should be considered in the management of the Taizi River catchment.

Four fish species, including *P. lagowskii*, *B. barbatula nuda*, *O. obscura* and *C. granoei*, that exhibit a preference for gravel sediment and cold clean water [11], are sensitive to urbanization and increased specific conductivity. In the current study, these species were only recorded at locations with low levels of anthropogenic disturbance and increased vegetation coverage. *O. obscura* in particular, was rarely recorded in the middle and lower reaches of the Taizi River, where there is a greater proportion of urban land use and higher specific conductivity values [55]. On the contrary, two fish species, *C. auratus* and *A. chankaensis*, were recorded at locations with a high proportion of urban land and high levels of specific conductivity. *C. auratus* is a common and cosmopolitan species in China, which is highly tolerant and adaptable [56]. *A. chankaensis* is a small pelagic species with a flat body. It is often distributed in the middle and lower reaches or city reaches of the river and shows a preference for deep water. These biological and ecological features and environmental tolerances showed congruency with the positive responses of *C. auratus* and *A. chankaensis* along the stressor gradient, with TITAN analysis in this study.

Sensitive species ($z-$) displayed a decline in frequency and abundance at a very low level of urban land cover (<3%, Figure 3a). The assemblage-level threshold of the proportion of urban areas in the catchment was 2.6–3.1% as determined by TITAN analysis (Table 5). These results are in keeping with the findings of Johnson [57] who reported that fish abundance can decline greatly when the proportion of urban land in a catchment exceeds 3%. In addition, declining fish integrity was noted when the proportion of urban land in the catchment was just 4% [58]. Despite this, Kovalenko et al. [59] found that different eco-regions had no fixed thresholds regarding the proportion of urban land, and change points in the community composition of sensitive aquatic taxa ($z-$) occurred at 4 to 6% cover of urban land in the catchment in the Northern ecoprovince, and 7 to 10% cover in the Southern ecoprovince of the US Great Lakes [53]. There are large regional differences in the patterns and development levels of urbanization between various countries and regions, making it difficult to obtain a uniform threshold for the proportion of urban land in a catchment. Determining a threshold for the proportion of urban land in the Taizi River catchment is extremely valuable for local catchment management, as biotic responses to urbanization occur at a very low proportion of urban land cover, and there is congruent sensitivity to urbanization among different biotic assemblages [59].

High levels of urbanization affect sensitive fish species, resulting in fish assemblages dominated by tolerant species [60]. To minimize the effects of urbanization on fish assemblages or sensitive species, attention should not only be placed on urban development, but should also focus on ecological restoration. For example, the creation or management of riparian buffer zones which reduce the hydrological connectivity between impervious areas and streams, reducing the negative effects of urbanization on fish assemblages [61].

Salinity in streams, routinely measured by conductivity, is increasing in many regions around the world and has changed aquatic communities [49]. Many countries have focused on conductivity in catchment management [62,63]. Hart et al. [64] suggested that freshwater fish can tolerate salinities of up to 10,000 mg/L (14,705 µS/cm). Other studies examined the salinity tolerances of various fish species, including yellow perch (*Perca flavescens*) [65], goldfish (*C. auratus*) [66] and carp (*Cyprinus carpio*) [67]. These studies were conducted under laboratory conditions and have not been compared with field data. In addition, very little attention has been paid to the specific conductivity thresholds for fish assemblages. As indicated by the results of the current study, the assemblage-level threshold of fish to specific conductivity ranged from 369.5–484.5 µS/cm. Zhao et al. [68] suggested that the assemblage-level threshold of macroinvertebrates to specific conductivity was 249 µS/cm in the Taizi River catchment. This threshold is slightly lower than the findings of the current research, as fish have stronger osmotic regulation capacities than macroinvertebrates [69]. Although specific conductivity is closely linked to geology, the significant increase in specific conductivity levels in the Taizi River over the last four decades (Table A2), highlights the necessity of a specific conductivity threshold for fish conservation.

Agriculture and deforestation contribute to the transport of silt into water bodies, often resulting in siltation of the river bed. This change in substrate results in a reduction of fish species with particular habitat needs, notably demersal feeding and spawning species [70]. Most of the sensitive species in the Taizi River prefer clean rock and gravel substrates, and are demersal feeders and spawners [11]. Functional metrics should be taken into consideration in future threshold identification studies, as they provide an interspecific perspective in understanding fish assemblages. Analyzing the underlying physiological principles is also of great importance in understanding the resilience and thresholds of fish species to environmental changes.

**Author Contributions:** All authors listed have contributed to this study. The analysis of the data and manuscript conception were done in cooperation with all authors. Y.Z. and S.D. wrote the manuscript.

**Funding:** This research was funded by the National Natural Science Foundation of China 41401066.

**Acknowledgments:** The authors wish to thank Xianwei Huang for providing excellent field assistance.

**Conflicts of Interest:** The authors declare no conflict of interest.

## Appendix A

**Table A1.** Water quality parameters and land use patterns in the Taizi River catchment (*n* = 53).

| Environmental Factors | Mean | Std. Dev. | Minimum | Maximum |
|---|---|---|---|---|
| Water physio-chemical conditions | | | | |
| WT (°C) | 21.11 | 3.19 | 14.8 | 26.8 |
| pH | 8.16 | 0.44 | 6.99 | 8.8 |
| DO (mg/L) | 6.8 | 1.57 | 3.9 | 13.5 |
| SC (µS/cm) | 377.49 | 210.2 | 74 | 1133 |
| TDS (mg/L) | 334.96 | 160.78 | 51 | 746.5 |
| SS (mg/L) | 158.1 | 162.25 | 8.5 | 884 |
| TN (mg/L) | 3.26 | 2.71 | 0.7 | 17 |
| TP (mg/L) | 0.14 | 0.13 | 0.03 | 0.6 |
| BOD (mg/L) | 5.48 | 4.38 | 1.9 | 28.7 |
| COD (mg/L) | 3.77 | 2.01 | 1.4 | 8.25 |
| $NH_4^+$ (mg/L) | 1.01 | 2.12 | 0.03 | 13.2 |
| Land use patterns | | | | |
| Forest (%) | 75.09 | 12.86 | 23.16 | 91.51 |
| Pasture (%) | 0.70 | 0.01 | 0 | 3.42 |
| Farmland (%) | 9.53 | 1.31 | 6.32 | 59.88 |
| Urban (%) | 3.80 | 0.52 | 0 | 21.01 |

**Table A2.** Historical data of average concentrations of the main ions in the Taizi River. Data obtained from the Hydrological Yearbook of the Liaohe River Basin, China.

| Ion | Year | | |
|---|---|---|---|
| | **1971** | **1980** | **2010** |
| Potassium + Sodium (mg/L) | 11.26 | 13.00 | 22.40 |
| Calcium (mg/L) | 16.09 | 24.59 | 39.06 |
| Magnesium (mg/L) | 5.45 | 6.37 | 12.96 |
| Sulfate (mg/L) | 13.68 | 22.71 | 70.71 |
| Chloride (mg/L) | 2.82 | 5.05 | 24.52 |

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
