# Peer review of "Threshold Responses in the Taxonomic and Functional Structure of Fish Assemblages to Land Use and Water Quality: A Case Study from the Taizi River"

_water, doi:10.3390/w11040661_

Round 1

Reviewer 1 Report

The paper by Zhang et al., titled "Threshold responses of fish taxonomic and functional structure to land use and water quality: a case study of Taizi River” (water-438562) aims to understand the relationships between fish taxonomic and functional structures and different human stressors by assigning threshold values of these drivers. The theme generally is interesting and might be innovative for Chinese River Basins, even though similar approaches were followed in the United States, Europe and elsewhere in the last decade.

However, the language is bad, important literature and methodological information is missing, results are confusing etc. Thus, my suggestion is to only accept this work, if major revisions are conducted. The English language will need some substantial editing. I provide a detailed list with general and specific improvements/edits to be made to the authors. I am willing to review a major revision of this manuscript, if done properly.

COMMENTS for Authors

GENERAL (concerns all chapters):

- I am irritated by the use of the terms “environmental drivers” and “environmental variables”, as the related variables (land use categories and water quality parameters) clearly are stressors (caused by humans) and not environmental variables (as e.g. biogeographical ones as altitude, air temperature etc.). Therefore I strongly recommend using the term “stressors” instead, in the entire document (and without “environment” in front).

- As currently written, the text is not well readable and the English language clearly needs substantial editing and revision by a native speaker. Be especially careful with pronouns, use/avoidance of article, singular/plural and word order (I give examples on how to improve ONLY for the introduction section, all other chapters are expected to re-vised by the authors).

- Moreover, there are sentences/paragraphs all over the document, which are confusing/hard to understand (they are mentioned in the respective sections below). Please re-write these in a clearer way and shorten sentences, where possible.

SPECIFIC (per chapter):

Abstract:

-      Line 12: The very first sentence of the abstract is confusing and seems to be incomplete. This also applies for the second sentence in the abstract. Please re-formulate according to my comments above.

-      Moreover, I’d like to see a stronger concluding statement, on how the paper’s results are innovative (and helpful for management).

-      Based on these comments and everything written below, please completely rewrite and improve the discussion section.

Introduction/discussion:

What I miss in the introduction and discussion in general are a few statements on the detailed impacts of land use and water quality stress on fish, both taxonomy and functionality (and related recent literature, see also specific comments below).

-      Line 35: Use “anthropogenic drivers” or just “drivers” instead of “environmental drivers”, see general comment above.

-      Line 40: “…above have detrimental effects on stream fish assemblages”.

-      Line 41: “…manager’s focus in river basin management”. (I suggest to also use the term “river basin management” where applicable in the entire document, as it is more specific).

-      Line 45: “…turning points, where the biological conditions above or below will alter significantly”.

-      Line 46: “Understanding the relationship…”.

-      Lines 55-57: Whole sentence “…to mitigate the potential risks”. Not clear what is meant here, please clarify and give a better explanation.

-      Lines 57-59: Statement “Although the functional traits...few studies…catchment land use” is simply not true. For example, Daniel et al. (2015) and Cooper et al. (2017) for the United States, or Logez et al. (2013), Schinegger et al. (2013) and Trautwein et al. (2013) etc. have examined such facts at large spatial scales, see

Daniel, W. M., Infante, D. M., Hughes, R. M., Tsang, Y. P., Esselman, P. C., Wieferich, D., ... & Taylor, W. W. (2015). Characterizing coal and mineral mines as a regional source of stress to stream fish assemblages. Ecological Indicators, 50, 50-61.
Cooper, A. R., Infante, D. M., Daniel, W. M., Wehrly, K. E., Wang, L., & Brenden, T. O. (2017). Assessment of dam effects on streams and fish assemblages of the conterminous USA. Science of the Total Environment586, 879-889.
Logez, M., Bady, P., Melcher, A., & Pont, D. (2013). A continental
scale analysis of fish assemblage functional structure in European rivers. Ecography36(1), 80-91.
Schinegger, R., Trautwein, C., & Schmutz, S. (2013). Pressure-specific and multiple pressure response of fish assemblages in European running waters. Limnologica-Ecology and Management of Inland Waters43(5), 348-361.
Trautwein, C., Schinegger, R., & Schmutz, S. (2013). Divergent reaction of fish metrics to human pressures in fish assemblage types in Europe. Hydrobiologia718(1), 207-220.

So please include this literature or other relevant one and correct your statement (maybe this is true for your basin or China in general?).

-      Lines 60-61: “…recent years in the light of global…”.

-      Line 62: “…requirements for natural resources…”.

-      Line 65:…”an awkward…”. This word should not be used in scientific English. I rather propose to use “…the limitations…”.

-      Lines 70-71: “…macroinvertebrates…, macrophytes…”.

-      Lines 75-76: I propose to state “…chose the Taizi River Basin as study area, as the tremendous changes in landscape patterns, water quality conditions and thus impacts on the aquatic community, especially fish assemblages. Understanding the drivers…”.

-      Line 80: “Our hypothesis was that the functional approach could identify specific drivers…”.

-      Line 83: “…specific drivers.”.

Methods:

-      Line 94: Not clear what is meant by an “obvious gradient of landuse patterns”, please clarify.

-      Line 104: Not clear why other 62 species should be occurring, according to which source? Give details, please!

-      Figure 1: The detail-image is not well visible, please increase figure size or improve in another way. Also, the landuse categories are not well visible in grey shades, I rather propose to use different colours here.

-      Line 115: It should be “properties” or even better “physical-chemical condition”.

-      Line 127: “300 ml…”. Never start a sentence with a number in a scientific publication.

-      Line 130: “Land use patterns…”.

-      Line 138: Citation/reference to version of ArcGIS are missing.

-      Line 139: Explain here, what is meant by upstream drainage area.

-      Lines 139-140: Not clear, how this reference to Figure 1 and Table A1 are fitting, please check/clarify.

-      Lines 141-156: Indicated which (standardized) methods for electric fishing etc. were used, why (especially if it concerns “non-wading electric fishing”, netting etc. and give references. Otherwise, catches and results are not correctly documented.

-      Line 149: Regarding unknown species and their formaldehyde treatment and take-out: Please give a statement on why there was no other option and address animal rights/ethics in this statement, according on how it is/should be handled in China.

-      Line 160: “…were Bonferroni adjusted…”. Please include a reference to the adjustment, too.

-      Line 176: Please add a reference on why Bootstrapping is a well-used method for indicator species identification.

-      Line 188: Citation/reference to version R are missing.

-      Table 1 and other: Check format of all tables, the layout is not ideal. Start with capital letters for all columns and rows.

Results:

In general, please check all sentences and paragraphs, sequence of words often is not ideal, especially here, English check should be done extensively.

-      Table 2: Bonferroni adjustment values is just too much. Instead, work with asterisks, as it usually is done to indicate level of significance (*, ** or ***).

-      Table 3: The text in brackets is unclear, way too long and maybe unnecessary. Please improve!

-      Figure 2: Not clear what is meant by “The same as below”. M1-M11 also could be shown beside the plots in the legend, with shorter names, e.g. “% pelagic species” etc., then the figure is more self-explaining.

-      Lines 250-253: Sentence “The species….urban proportion” needs to be reconsidered and rewritten.

-      Line 265: “Species and assemblage thresholds for stressor variables”.

-      Line 270 and 272: Explain what is meant by negative/positive indicator species already here.

-      All further parts of results: Please carefully revise according to all my other comments before.

Discussion:

General: According to all my statements before, please completely rewrite and improve the discussion section.

Author Response

Dear reviewer:

Thank you very much for your valuable comments on our manuscript. According to your comments, the manuscript has been edited by International Science Editing.

Reviewer 2 Report

Overall the paper was interesting and fits an area in need of study by examining threshold effects instead of simply looking at correlations with LULC data.  I found the paper easy to read and the data easy to follow.

Overall general comment:  I have one major concern that needs to be addressed prior to acceptance that prevented me from adequately evaluating the discussion.  The authors failed to take into account drainage size when looking at the habitat and land use variables.  It is likely that many of these vary along a longitudinal gradient and without taking this into account (starting possibly with partial correlations) it is hard to determine if fishes are responding to water quality and land use and not simply stream size.  This could have influenced most of the variables examined in the study and therefore could lead to overinflation of the importance of environmental data as drivers for changes in the fish assemblage and species occurrence across sites.  Most of the forest sites were in upland, mountainous streams while the impacted sites were all in areas with less gradient. I also can't tell from the map for sure but it appears that more of the tributary sites are located in the forested, upland region. If the authors could redo the analyses taking into account catchment size I believe it would remove this concern and provide a more valid comparison across sites.  

Specific comments:

L 139-140:  Was the entire upstream drainage area examined for LULC or was a buffer applied?

L141-156: Please provide more details on the electroshocking methods and define microhabitat.  Were dip nets used and was this standardized?  How many dip netters were there?  Did stream width variety considerably and how did this influence sampling?  How were fishes classified to environmental disturbance (which disturbances were taking into account)? Did all sites have riffle, run, pool morphology?  

L157: I strongly encourage you to consider the impact of catchment size on the land use and physio-chemical variables measured in the study.

L171-75: Justify your use of raw abundance data across such a wide variety of sites and sampling conditions instead of relative abundances?  I would anticipate that raw abundances would vary simply as a result of sampling efficiency at sites and could therefore be strongly biased by in stream habitat at a site.  It appears that sampling was only roughly standardized (1 hr) but not reach length, steam size or use of block nets.

Are variables like water temperature correlated with forest simply due to catchment size?  Are these systems spring fed?  This is one example of how catchment might influence your variables.

Author Response

Dear reviewer:

Thank you very much for your valuable comments on our manuscript.  The manuscript has been edited by International Science Editing.

Reviewer 3 Report

The study aimed to show the importance of functional traits to identify specific drivers and their thresholds related to fish stream assemblage at the catchment scale. In general, the manuscript is well written, understandable, but only few minor things should be changed or revised (see below). Otherwise, I am satisfied with the manuscript.

Lines

26 what do you mean by a prior sensitive response

65 an awkward is not noun - rewrite it

121 determined at left, middle and right of the channel

127 Not appropriate to begin a sentence with a digit

132 why were images not from the years of the study sampling? Availability?

154 how were sensitive and tolerant species determined exactly? Could you specify it or at least provide a reference?

196 a total of 5,342 fish (comma)

215 Table 2 – I would remove (ns) from the table, making it unreadable, or unclear. It is customary to mark only significant results. What does (0.000) mean? When the level is more than 0.001, then 10-6. Or better specify.

223 mismatch between farmland and NH4 p-levels in the text and Table 3

233 aixs1 and aixs2 to axis1 and axis2

242 Figure 2. Why did you use M labels for naming fish guilds? Why don’t you use same labelling as in Table 1 (instead of M4, Omni for an omnivorous group, etc.)? It would be clearer to the reader than always read it from the figure caption for what fish group M-whatever stands for. The graph and also text would be more straight forward and comprehensive. In addition, I would go for a group or maybe guild, but not for proportion of individuals as whatever (e.g. Pelagic species group, or just Piscivorous group), easing the reading and comprehensiveness.

258 Here you used only whole names of species group (proportion of individuals as …) without M-labels, but further supplemented with M-labels (296), and even further used only M-labels (313). Please, unify this.

265 please, switch the order of urban and SC in the text or better in Figure 3. Figure 3a should be urban and Figure 3b should be SC.

313 what did you mean by a prior threshold?

318 Figure 5 caption – should include a basic description of box lines and points

323 what did you mean by a nested effect and two-scale stressors

324 attentions -> attention

350 had strong disturbed -> rewrite it

355 more attention has been

357 how do you mean that functional structure is a tangible indicator?

392 attentions -> attention

399 10,000

432 it is very peculiar that who wrote the manuscript is not the first author. Why? Just curious.

Author Response

(The authors gave the same response as above.)

Reviewer 4 Report

General comments:

 This a very interesting work that tackles fish taxonomic and functional responses to land use and water quality changes with appropriate methods. The work is well written and well executed. I commend the authors for their work. I have some minor comments bellow that I hope can contribute to improve the work.

Specific comments:

Lines 94-96 – Please provide references for this information

Line 143 – Please describe the electrofishing sampling procedure.

Line 150 – How were the fish identified? Please provide a description and references for the guides or nomenclature keys.

Line 167 – Why use a VIF of 20 as a cut-off point? Some authors suggest 2 or 3 or even other values. Please provide justification ad reference.

Lines 178-181- Please provide references for this.

Figure 2 – Part of the labels of the vertical axe are cut.

Figures 2, 3 and 4 – The captions are not fully descriptive. They redirect to other captions, but I would suggest full clarity in each caption.

Author Response

(The authors gave the same response as above.)

Round 2

Reviewer 1 Report

Through your major revision, you did a good job. However, I still suggest minor revisions on specific aspects listed below:

Abstract, Line 33: I would like to see a more engaging and active statement here, not only that it should not be ignored, i.e. that the found facts are very important aspects for future catchment management etc.

Line 40: influences --> influence

Line 41: affects --> affect

Line 50: I am still irritated by "environmental drivers".

Line 52: Same here, "environmental gradients" to me means that e.g. altitude, slope etc. are concerned, but not human stressors. So I strongly recommend to only use the term "drivers" here, or "anthropogenic drivers".

Line 63: As stated before, I strongly recommend to at least cite "Daniel et al. (2015)", as you should indicate related studies you know of.

Line 67: If you want to proceed with "environmental thresholds", I still recommend to clarify once, what you mean with it here in this study, i.e. tipping points of fish functional aspects to human stressors.

Line 69: Same applies for "environmental gradient". Otherwise, it implies you mean the ranges of altitude and other environmental/abiotic variables.

Line 78: Please state "Taizi River catchment"  or "Taizi River basin" (also elsewhere in the manuscript).

Line 80 (and others): "environmental drivers" (see comment above).

Line 110: Here I would add "at local and catchment scale".

Line 161: significant --> significance

Lines 318-320: Font size changes!

Line 323: I suggest to re-write: "that it is essential to focus on efforts towards an integrated management...".

Line 330: contributes --> contribute

Line 332: affects --> affect

Line 333: affect --> affecting

Line 337: "environmental stressors" --> see comments above.

Line 342: I would state "...in changes of functional...".

Line 364: were reported --> was reported

Line 389: affects --> affect

Lines 412-413: I suggest "...fish species is easier to understand for managers...".

Line 415: strongly --> stronger

Line 421:  I suggest "...resulting in siltation of the river bed..".

Author Response

Dear reviewer

Thank you very much for your valuable comments on our manuscript.

Reviewer 2 Report

I would have preferred to see data that support what you are saying because it is simply not clear looking at the map provided (which indicates there may be a significant difference in drainage size between forested and urban sites.  There are numerous papers that show the change in fish assemblages along a longitudinal gradient. You may be correct and it is not a factor in their design but you have failed to provide supporting evidence for this assertion. I still am concerned that catchment size is an issue as well as standard sampling effort across sites and assuming that total abundance is adequate (as opposed to using relative abundance).

Author Response

(The authors gave the same response as above.)

Round 3

Reviewer 2 Report

I approve acceptance of the manuscript after suggested revisions were considered.